# Mitochondrial Dysfunction in Alzheimer’s Disease: A Biomarker of the Future?

**DOI:** 10.3390/biomedicines9010063

**Published:** 2021-01-11

**Authors:** Simon M. Bell, Katy Barnes, Matteo De Marco, Pamela J. Shaw, Laura Ferraiuolo, Daniel J. Blackburn, Annalena Venneri, Heather Mortiboys

**Affiliations:** Sheffield Institute for Translational Neuroscience (SITraN), University of Sheffield, 385a Glossop Road, Sheffield S10 2HQ, UK; kabarnes1@sheffield.ac.uk (K.B.); m.demarco@sheffield.ac.uk (M.D.M.); pamela.shaw@sheffield.ac.uk (P.J.S.); l.ferraiuolo@shef.ac.uk (L.F.); d.blackburn@sheffield.ac.uk (D.J.B.); a.venneri@sheffield.ac.uk (A.V.); h.mortiboys@sheffield.ac.uk (H.M.)

**Keywords:** mitochondria, Alzheimer’s disease, biomarker

## Abstract

Alzheimer’s disease (AD) is the most common cause of dementia worldwide and is characterised pathologically by the accumulation of amyloid beta and tau protein aggregates. Currently, there are no approved disease modifying therapies for clearance of either of these proteins from the brain of people with AD. As well as abnormalities in protein aggregation, other pathological changes are seen in this condition. The function of mitochondria in both the nervous system and rest of the body is altered early in this disease, and both amyloid and tau have detrimental effects on mitochondrial function. In this review article, we describe how the function and structure of mitochondria change in AD. This review summarises current imaging techniques that use surrogate markers of mitochondrial function in both research and clinical practice, but also how mitochondrial functions such as ATP production, calcium homeostasis, mitophagy and reactive oxygen species production are affected in AD mitochondria. The evidence reviewed suggests that the measurement of mitochondrial function may be developed into a future biomarker for early AD. Further work with larger cohorts of patients is needed before mitochondrial functional biomarkers are ready for clinical use.

## 1. Introduction

Cellular metabolic changes within the brains of people with Alzheimer’s disease (AD) are seen very early in the condition, and often precede the development of both amyloid plaques [1,2,3,4] and neurofibrillary tangles [5,6,7,8]. Abnormalities have been shown in many metabolic pathways in AD [9], with both peripheral and nervous system cells affected. Mounting evidence suggests that deficits in the function of mitochondria, specifically how they control oxidative phosphorylation (OxPHOS) [10], are likely to be key in the development and progression of AD. In fact, an alternative mitochondrial hypothesis for the aetiology of AD states that people who inherit mitochondrial genes that predispose them to lower mitochondrial respiration rates are more likely to develop the condition [11]. A full understanding of the mitochondrial abnormalities identified in AD may lead to new drug treatments or biomarkers to treat and detect the disease. 

Mitochondria are double membraned organelles abundant in almost all types of mammalian cells [12]. Mitochondria are in a constant state of flux, altering morphology and localization depending on energy demands or metabolic stresses within the cell [13]. The main cellular function that is performed by the mitochondria is the production of ATP via the electron transport chain (ETC). Complexes I–IV of the respiratory chain are coupled to the action of the F0F1-ATP synthase (Complex V) enzyme, which uses the membrane potential generated by Complexes I–IV to generate ATP from ADP and phosphate [14]. In addition, mitochondria are critical to many other cellular functions, including maintaining cellular calcium concentrations [15], the generation of reactive oxygen species (ROS) for cellular signalling and as a consequence of the inefficiency of the ETC [16,17,18,19]. Mitochondria also have roles in steroid synthesis, hormone synthesis and apoptotic signalling [20,21,22]. 

Mitochondria exist in a dynamic network, altering shape in response to stress or as a result of the metabolic demands of the cell. Mitochondria will often fuse together in times of increased energy demand, or metabolic stress (stress-induced mitochondrial hyperfusion) [12]. Mitochondrial fission is less directly linked to managing the ATP demands of the cell, but is used as a way of identifying defective mitochondria that need to be removed and recycled [23]. Mitophagy is the specific form of autophagy in which mitochondria are targeted and undergo degradation. Figure 1 highlights the functional properties of mitochondria.

Evidence suggests that many of the above functions of mitochondria are altered in AD [24,25,26,27]. The aim of this review is to present the evidence for altered mitochondrial function in AD in both the nervous system and peripheral cells. As mitochondrial dysfunction can be detected early in the course of AD, it has the potential to be developed into a future biomarker for the condition. A body of literature already exists which describes how mitochondrial functional properties are utilised in imaging studies to identify metabolic alterations associated with AD. This review will finish with a discussion of some of the already used clinical imaging applications of mitochondrial dysfunction in AD, and what further steps need to be taken to develop the measurement of mitochondrial function into an AD biomarker of the future.

## 2. Electron Transport Chain Disruption in AD

Much of our knowledge about mitochondrial dysfunction in the human tissue of AD patients comes from post-mortem brain samples, of which the vast majority of studies are in patients with sporadic AD. Studies investigating the expression of OxPHOS proteins and mitochondrial DNA (mtDNA) have revealed potentially conflicting results. A micro-array analysis of post-mortem frozen hippocampal samples has revealed a global decrease in nuclear encoded OxPHOS protein subunits and no change in mitochondrial DNA (mtDNA) encoded subunits when AD brains are compared to both aged-matched controls and patients with MCI [28]. A study investigating mRNA levels in the mid temporal gyrus has revealed a decrease in the mRNA that encodes subunits MTCO1, and MTCO2, of Complex IV within the mid-temporal lobe [29]. A similar reduction in RNA of subunit MTCO3 of Complex IV has also been shown in the same brain areas [30]. These subunits are all mitochondrially encoded [31], but do not appear to be associated with a loss in mitochondria number, or correlate with amyloid brain levels [32,33]. Subunit MTCO2 RNA of complex IV has also been shown to be decreased in the hippocampus of the AD brain, but in the same study, no change was seen in the RNA of nuclear encoded subunits of Complex IV [33]. In contrast, both total cellular mtDNA and Complex IV protein levels in the AD hippocampus, frontal and temporal lobes have been found to be increased in AD [34]. This latter study reports that the majority of the mtDNA and Complex IV protein is not found within the mitochondria but in the cytoplasm, suggesting a greater turnover of mitochondria or a decrease in their proteolytic breakdown [34]. This study reports no difference in mtDNA expression in glial cells or neurons in areas of the brain not classically affected by AD. Further studies have suggested that the expression of mitochondrially encoded subunits of Complexes III and IV are increased in the AD brain, whereas mitochondrially encoded subunits for Complex I are decreased [35,36]. Together, these reports suggest that the turnover of mitochondria is altered in the AD brain, with the potential consequence of alterations in the expression of ETC proteins.

The apparent conflicting nature of the results across the papers described above may be explained by the fact that each study has small numbers of participants and uses different techniques to preserve the brain samples used for analysis. RNA is known to be unstable when prepared from preserved tissue samples, and this feature may be a source of conflicting results. It has been shown that the AD brain has a higher load of both mtDNA and nuclear DNA mutations, a finding that may also help to explain the differences in results seen across studies [37,38,39]. Changes seen in these studies are not specific to AD, with similar changes also seen in the brains of people with autism [40]. The fact that differing expression profiles of the ETC RNA and protein subunits differ between these studies may also highlight the heterogeneity of AD, and explain how, in sporadic AD, there are likely to be multiple factors that affect the expression of ETC proteins. These studies focus on cases that have high disease burden, and therefore changes may not be triggers of AD, but more consequences of the progression of the disease. The changes seen in post-mortem studies may be a consequence of disease progression and may not be present at the start of disease. This would potentially make these changes unsuitable for biomarker and therapeutic development. 

The functional assessment of Complex IV from post-mortem tissue has shown decreases in activity in brains from sporadic AD patients [41,42,43,44,45,46]. The frontal, temporal and parietal lobes in certain studies show a selective reduction in the activity of Complex IV [41,42], but this finding is not consistent across all studies, with studies using very similar techniques showing deficits in Complex IV activity throughout the whole brain [43,44,46]. The reason for the reduction in Complex IV activity seen in the AD brain is not known, but the structure of the complex IV protein is changed in AD, potentially as a result of oxidative stress [45]. The loss of neurons seen as AD progresses has also been suggested as a cause for the apparent reduction in Complex IV activity [46]. This theory is corroborated by evidence from animal models, showing Complex IV activity and protein reduction as a consequence of loss of neuronal activation from the application of tetrodotoxin [47,48,49]. A loss of neuron/synaptic number, though, would not fully account for the selective loss of activity of complex IV reported in these studies, although it puts forward the hypothesis that OxPHOS demand may modulate ETC activity and protein expression [50]. Interestingly, the activity of the other ETC has been reported to be reduced in AD post-mortem brain samples [43,44], but this finding is not repeated across all studies. Through post-mortem brain samples, it is difficult to ascertain if the changes seen in Complex IV activity are present from birth or are the consequence of another pathological process involved in AD, such as protein aggregation, or synapse loss. 

The study of blood cells, almost exclusively in sporadic AD patients, has revealed functional changes in the OxPHOS pathway. Peripheral blood mononuclear cells (PBMCs) show decreased basal oxygen consumption rates (OCR) and proton leak in AD, but no change in mitochondrial maximum respiratory capacity when compared to age matched controls [51]. This work is complemented by work specifically in lymphocytes (a type of PBMC), which shows a reduction in basal OCR, but also a reduction in maximum respiratory capacity of lymphocyte mitochondria [52]. Lymphoblastic cell lines (LCL) from AD and Down’s syndrome (DS) patients have a low mitochondrial membrane potential (MMP) in both old and young DS patients, but ATP loss was not seen until later in the disease, when AD developed [53]. This suggests that the loss of ATP maintenance is developed through the course of AD as opposed to blood cells having a lower level of cellular ATP prior to disease onset. In whole blood samples from people with AD and mild cognitive impairment (MCI), the OxPHOS genes that are nuclear encoded are downregulated and those that are mitochondrially encoded are unregulated [54]. It is unclear if this affects the function of the ETC, but it is interesting that differences in the expression of mtDNA genes are seen between peripheral and CNS tissue samples. This observation may suggest that different mechanisms are present in a particular cell type to combat the effects of altered OxPHOS gene expression and may go some way to explaining why AD does not manifest itself in non-CNS tissues. These changes in OxPHOS seen in AD blood cells are likely to be the cause of the decreased MMP and reduction in total cellular ATP levels that have also been reported in platelets, and PBMCs [52,55].

Platelets have also been shown to have abnormalities in the ETC in AD, and are often suggested to be a good peripheral cell model for the disease [56]. Platelets contain the enzymatic pathways needed to produce Aβ from the APP protein, and have been shown to secrete Aβ into the blood stream [57,58]. AD patient platelet OCR is similar to controls until platelets become activated through exposure to substances such as collagen, thromboxane or monoamine neurotransmitters [58,59]. Reductions in basal and maximum OxPHOS capacity are thought to be due to a combination of reductions in complex I substrates, lower activities of complex IV, increased activities of complex I and reduced concentrations of ubiquitin [55]. The evidence that platelets only start to show deficits in ETC function when activated suggests capacity deficits in the ETC function in AD that may only be measurable at times of physiological stress. Several studies have shown both a decrease in the activity and the expression of the Complex IV enzyme and its subunits in the platelets of patients with AD [59,60,61]. The changes seen in the metabolism of blood cells in AD could be an example of inherent deficits in mitochondrial function, but some evidence from animal models suggests that the changes in blood cell ETC function may be precipitated by the development of the pathological aggregates of AD within the brain [59]. A pro-inflammatory environment within the brain is established that then activates platelets leading to the OxPHOS deficits [59]. The changes in metabolic function seen in PMBCs may also be a direct consequence of the changes seen within the brain, but this has not yet been shown. 

Fibroblasts from patients with both sporadic and familial AD have been extensively studied for both functional and structural abnormalities in AD. Like platelets, fibroblasts can produce Aβ, and this production is increased in fibroblasts from patients with AD [33,62]. The function of the ETC of fibroblasts from patients with AD shows more variable results than those from blood cells. ATP and MMP levels have been shown to be low in some studies of both sporadic and familial AD fibroblasts [63,64,65,66]. However, normal and higher than control levels of ATP and MMP have also been found in AD fibroblasts [67,68,69]. Alterations in the NAD+/NADH ratio, and how this is maintained, is seen in fibroblasts from patients with sporadic AD who have high OCR and MMP compared to controls [68]. This suggests that the functional changes seen in the mitochondria of AD fibroblasts may not be a direct consequence of ETC activity, but caused by the substrates that interact with the different complexes. This paper also highlights how glucose uptake appears to be impaired in AD fibroblasts, which again would affect ETC substrate availability [68]. The design of each of these studies on AD fibroblasts is quite different, with control group comparators ranging from being completely disease free to having other forms of dementia and neurological illness. Culture media differ between studies, with some using low glucose media and others using high. This will obviously affect mitochondrial function as substrate availability is different. The passage used between studies also varies, and as cultured human fibroblasts are affected by aging, this could also affect metabolic function. This may explain why fibroblast studies reach less of a consensus about the functional mitochondrial alterations seen in AD. All these studies agree that there are deficits in the function of the ETC, which may not necessarily affect mitochondrial function until a stressor is added [67]. As with post-mortem samples, and blood cells, inhibition of the Complex IV enzyme in fibroblasts is also stipulated to be the cause of the changes in ETC function [67]. 

It remains unclear if the functional and structural changes to the ETC seen in AD are a result of a primary mitochondrial change, as suggested in the mitochondrial hypothesis of AD [70], or a consequence of the build-up of amyloid within the brain and body of AD patients. Evidence from animal and cell models suggests that both amyloid and tau have a direct effect on the function of the ETC. In triple transgenic 3xTg-AD mice (human APPSWE, TauP301L, and PS1M146V genes) abnormalities in mitochondrial function are seen in the embryonic stage and in young mice long before the build-up of amyloid [71]. In this AD mouse model, most of the subunits of Complexes I and IV are downregulated and Complexes III and V are upregulated when the mitochondria are isolated and examined at 6 months [72]. Interestingly, in the APP23 mouse, the upregulation of both glycolysis and OxPHOS is seen before amyloid deposition, and this seems to increase the oxidative stress within cells [73]. This appears to be opposite to the metabolic changes that are seen in the transgenic 3xTg-AD mice. PSEN2 plays a role in maintaining MMP, as knock down of PSEN2 in mouse embryos reduces MMP [74,75], showing that two of the key mutations which cause familial AD may have a link to the altered metabolism seen in AD models. Other studies using triple transgenic mice that have mutations in the APP gene and an increased propensity to develop tau pathology (human APPSWE, TauP301L, and PS1NI41) have shown deficits in MMP, total cellular ATP, mitochondrial spare respiratory capacity (MSRC) and OCR. Rhein et al. 2009 have shown that tau preferentially disrupts complex I of the ETC, whereas amyloid preferentially disrupts Complex IV [76]. The effect tau has on Complex I has also been highlighted in studies looking at the effects of the plant poison annonacin, which has a structure very similar to tau and can also cause Complex I deficits [77]. This study shows that reduced Complex I activity leads to a redistribution of tau towards the cell stoma, which can potentiate the development of NFT and cell death, suggesting that tau pathology can also be exacerbated by poor mitochondrial function. The tau and amyloid effects on the ETC have also been shown to work synergistically to increase the speed at which mitochondrial dysfunction happens in animal models [78]. It has been shown that the effects of tau on the function of the ETC are propagated by the addition of amyloid [79]. 

Multiple cell line experiments have shown that overexpression of the APP protein affects the activity and structure of mitochondria. APP overexpression has revealed that Complexes I, II and IV all have reduced expression in the presence of Aβ [80,81], leading to reduced MMP and ATP production. The trafficking around the cell and structure of mitochondria is also affected by Aβ [26], with mitochondria shown to be more fragmented and to have a reduced MMP [27]. Removal of the mitochondria from a cell line can stop the toxic effect of Aβ on the cell line [81], again linking metabolism to Aβ pathology.

Limited evidence for a definitive primary mitochondrial pathology has been identified so far in AD, but a recent study has shown that a point mutation in the PTCD1 protein, encoded by nuclear DNA, is much more prevalent in people with AD [82]. This protein is essential for the normal assembly of mitochondria and this mutation has been shown to cause deficits in OxPHOS. Interestingly, cybrids (SH-SY5Y cell line with mtDNA removed and human platelet mtDNA added) created from patients with AD and controls also reveal reductions in the activities of Complexes I, II and IV, an increase in reactive oxygen species production, and a 40–50% reduction in ATP generation [83]. Changes in mitochondrial structure, including shorter length, increased fragmentation, and a collapse of the mitochondrial network were also seen. This is without the presence of APP, suggesting that the Aβ protein may exacerbate a mitochondrial phenotype that already exists in AD.

Research into mitochondrial ETC deficits seen in AD reveals mixed results depending on the situation in which the mitochondria are observed or the tissue in which they are studied (e.g., post-mortem brain, peripheral blood cells in people living with AD or animal models). Interestingly, the changes to ETC gene expression in peripheral cells are often associated with a reduction in ATP production, suggesting a reliance on OxPHOS for ATP production in the periphery. Inconsistency in findings could be explained by differences in experimental techniques, or the dynamic nature of mitochondria. Constituents of media, such as the concentration of metabolic substrates, can affect mitochondrial function and may explain differences among studies. Studies investigating the function of the ETC in sporadic AD rarely separate samples based of ApoE status (the greatest genetic risk factor for developing sporadic AD), a methodological shortcoming that may also contribute to conflicting results. The causes of ETC function and protein expression are very likely to be different between sporadic and familial AD, as the genes that become mutated in familial AD have functions within the mitochondria, independent of amyloid production. An example of this is in the maintenance of calcium balance, which is discussed below.

## 3. Mitochondrial Dynamic Changes Seen in AD

In normal conditions, during mitochondrial fusion, the fusion of the outer mitochondrial membranes is mediated by mitofusin 1 and 2 (Mfn1) and (Mfn2), whereas the fusion of the inner mitochondrial membrane is mediated by Optic atrophy 1 (OPA1). During mitochondrial fission, dynamin-related protein 1 (Drp1) is recruited to the mitochondria by four receptors on the outer mitochondrial membrane, fission 1 (Fis1), mitochondrial fission factor (Mff) and mitochondrial dynamics proteins of 49kDa and 51 kDa in size (MiD49/MiD51). Drp1 then forms an oligomeric ring structure around the mitochondrion which constricts, dividing the mitochondrion into two. The post-translational modification of Drp1 is known to be highly important in the action of Drp1. For example, the translocation of Drp1 is reliant on the phosphorylation state, Drp1 must be dephosphorylated at ser637 and phosphorylated at ser616 [84]. 

The fusion proteins OPA1 and Mfn2 have been found to be increased in hippocampal tissue taken from an APPsw/PS1dE9 mouse model by 12 months [85], though Calkins et al. 2011 [86] saw a decrease in Mfn1 and Mfn2 levels in primary neurons from a Tg2576 mouse and Trushina et al. 2012 [24] saw no change in either APP, PSEN1 or APP/PSEN1 mice. Changes to Mfn2 levels can be affected by age and sex in 3xTg mice, with reduced Mfn2 levels noted in cortical mitochondria at 6 and 14 months in both male and female mice, but only in males at 2 months [87]. Decreased OPA1 has been noted in the M17 human neuroblastoma line, overexpressing wild type APP [27], although OPA1 and the mitofusins were found to be increased in HEK293 cells, an embryonic kidney cell line, when human tau was overexpressed [88]. Reduced levels of several OPA1 isoforms have been seen in sporadic patient-derived fibroblasts, with an increase seen in a particular short isoform [64], though this is not consistent, as some have seen no change [66,89]. To date, few studies have investigated these proteins in a disease relevant, patient-derived model, though Birnbaum et al. 2018 [90] used iPSC-derived neurons from five sporadic AD patients and two controls, and saw no change in Mfn1 or Mfn2. 

Fission proteins have also been studied, in particular Drp1. Increased Drp1 has been seen in both transgenic [85,91,92] and streptozotocin-induced [93] mouse models of AD, although this was sex-dependent [87]. An increase in Drp1 was seen in cortical and hippocampal mitochondria from 3xTg mice at 2 and 6 months in females only, whilst males showed reduced Drp1 levels in cortical mitochondria at 6 months and no significant difference in hippocampal mitochondria. Decreased levels of Drp1 have also been noted in the M17 neuroblastoma line overexpressing wild type APP [27], and post-mortem tissue [94], as well as in both sporadic and familial AD patient fibroblasts [27,66,95], though this is not consistent [91]. In sporadic AD iPSC-derived neurons, no change was seen [90], whereas astrocytes expressing ApoE4, a significant risk factor for sporadic AD, showed decreased levels of Drp1 [92]. Mitochondria-localised Drp1 has also been seen to be reduced in sporadic AD [66,95,96], which may be due to the reduced overall levels of Drp1, or may suggest an impairment in the recruitment of Drp1 to the mitochondria.

Fis1 has been seen to be increased in various models of AD, both where Drp1 was seen to be increased [97,98], and interestingly, where Drp1 was seen to be decreased [89,94]. The importance of Drp1 post-translational modification may somewhat explain these potentially conflicting results. Whilst many studies focus on the mRNA or protein levels of Drp1 and Fis1 in AD, Joshi et al. (2018) investigated the interactions between Drp1 and Fis1, using an inhibitor of these interactions, P110. They found that treatment with P110 prevented alterations to mitochondrial morphology and function in a range of models of AD, including sporadic patient fibroblasts, transgenic mice and cultured neurons. 

As previously mentioned, post-translational modification is key in the action of Drp1. S-nitrosylation, the covalent binding of a nitric oxide group to a cysteine thiol to form an S-nitrosothiol (SNO), may be of particular significance in AD. Nitric oxide (NO) is produced in response to Aβ, leading to the s-nitrosylation of Drp1 to SNO-Drp1. This leads to increased mitochondrial fission, loss of synapses and neuronal damage in both rat primary neuron cultures [95] and post-mortem tissue [99]. SNO-Drp1 has been found to be increased in peripheral blood lymphocytes of AD patients [96], as well as post-mortem AD patient brains, and preventing nitrosylation has been shown to attenuate neuronal damage [99]. However, others have found that s-nitrosylation has no effect on Drp1 activity, and SNO-Drp1 is not significantly different in AD compared to controls [100].

Mitochondrial biogenesis is also altered in AD. Mitochondrial biogenesis refers to the generation of new mitochondria from the growth and division of an existing mitochondrion [101]. Studies have shown the master regulator of mitochondrial biogenesis, peroxisome proliferator activator receptor gamma-coactivator 1α (PGC-1α), is downregulated at the mRNA and protein level, specifically in hippocampal regions of the AD brain rather than the cerebellum. Furthermore, the same reductions have been found in several downstream regulators, including nuclear factor erythroid 2-related factor (Nrf2), Nrf1 and TFAM in the same brain region of the AD brain [102,103]. These reductions have also been found in animal and cellular models of familial AD, showing that this is a pathway which is dysregulated in both sporadic and familial disease [103,104,105]. This suggests a reduction in the ability of a cell to create new mitochondria in AD. It has also been shown that Aβ-induced cellular death can be reduced by upregulation of Nrf2, which is thought to be mediated by both increased oxidative stress resistance and improved mitochondrial biogenesis [106,107]. Interestingly, Aβ has been shown to have a direct effect on mitochondrial biogenesis in the cell models of AD, which can be corrected with the application of both drugs, such as diabetic medication pioglitazone and naturally occurring compounds like curcumin [108,109,110,111,112,113,114].

## 4. Mitochondrial Calcium Signalling in AD

The tight control of both intracellular calcium and intramitochondrial calcium has been shown to be affected in AD. PM samples from AD patient frontal cortex tissue show that the Na/Ca exchanger that maintains mitochondria matrix calcium is remodelled, and has reduced expression [15]. This leads to an increase in matrix calcium concentration. In the same paper, a transgenic mouse (3xTg-AD) with the Na/Ca exchanger deleted is shown to have an accelerated amyloid and tau pathology, as well as increased memory loss. The cognitive deficit seen in the mouse model can be corrected by the normal expression of the Na/Ca exchanger, suggesting that high levels of mitochondrial matrix calcium are essential to the development of the pathology of AD. Aβ has also been shown to increase intracellular calcium levels, which can increase cellular vulnerability to excitotoxicity [115]. This will also have an effect on mitochondrial function as the calcium buffering capacity of the mitochondria is challenged by an increased intracellular calcium. A recent study using an animal model of AD (the APPswe/PSEN1∆E9 (APP/PS1) Tg mouse) has suggested that Aβ increases mitochondrial calcium levels via its actions on the mitochondrial calcium uniporter [116]. The same study shows that in the post-mortem AD brain, the expression of influx calcium transporters is reduced, while efflux transporters are increased, suggesting an adaptive mechanism by neuronal cells to manage the increased mitochondrial calcium load caused by Aβ. The efflux of calcium from mitochondria has been shown to be further impaired by the presence of misfolded tau [117,118], giving further evidence of the synergistic effect that both tau and amyloid aggregates have on the function of mitochondria in AD. 

The buffering of calcium by the mitochondria and endoplasmic reticulum is particularly affected by mutations in the proteins associated with familial AD (PSEN 1 and 2, and APP genes) [119,120,121]. The presenilin proteins have a role in regulating endoplasmic reticulum (ER) calcium release and maintaining cytosolic calcium concentrations [122]. The presenilin-mediated release of calcium has been shown to have a direct effect on reducing MMP, which in turn, can predispose cells to increased autophagy [119]. The presenilin-1 and -2 proteins are enriched in a specialized part of the ER called the mitochondrial associated ER membrane (MAM). The MAM sites are where the mitochondria and ER directly interact, and as they are enriched in presenilin proteins, this may explain why mitochondria accumulate Aβ and why presenilin-mediated calcium release leads to reduced MMP [123]. In cell models, the application of Aβ to neurons increases mitochondrial calcium levels via upregulating the inositol-1,4,5-triphosphate receptor-voltage-dependent anion channel (IP3R3-VDAC) contacts, also part of MAMs [124]. This same study showed that the IP3R3-VDAC contacts are increased in post-mortem AD brain samples, although the article did not state if the post-mortem samples come from patients with a familial or sporadic form of the disease. A research consensus, though, does not exist on the effect that familial AD associated protein mutations have on MAM contacts, with both increases [125] and decreases [126,127] in MAM contacts reported in several different model systems. 

The fact that presenilin proteins are a key part of the MAMs highlights what is very likely to be an important difference in what determines the pathogenesis of the mitochondrial dysfunction in familial and sporadic forms of AD. Current literature appears to suggest that the abnormal mitochondrial calcium concentration and signalling seen in AD is a direct effect of the Aβ and Tau proteins or caused by the presence of mutations with the PSEN1 and 2 genes associated with familial AD. Evidence of an inherent change in calcium concentration, or signalling abnormality is yet to be found, although since most models that investigate calcium signalling in AD rely on the pre-exposure to Aβ, then inherent calcium changes would be difficult to identify. It has been shown that MMP is reduced in both sporadic and familial AD patient cells; therefore, since mitochondrial calcium concentration is dependent on MMP, this may indicate that mitochondrial calcium concentrations are affected independently of the effect of amyloid and tau. 

## 5. Mitochondrial ROS Production in AD

Mitochondria are the main source of ROS production within a cell, accounting for 90% of all ROS production [128]. ROS are used by cells as signalling molecules, but in AD, the tight balance between the production of ROS and breakdown is altered. As ROS molecules are difficult to study directly, often evidence of ROS activity is identified via the oxidization of biological molecules. In AD, there is evidence of increased lipid, protein, DNA and RNA oxidation, both centrally [16,17,18,19] and peripherally [129,130], suggesting an increase in ROS production associated with the disease. As a result of increased oxidation, cell physiology in AD is put under strain, as the oxidized molecules either cease to function or develop abnormal function. 

Studying patients with DS has shown that oxidative stress in AD is an early event. The brains of patients with DS develop oxidative damage years prior to the build-up of the amyloid and tau aggregates [131,132]. This finding is also replicated in animal and cell models of AD that show increased oxidative damage prior to amyloid deposition [27,133,134]. In human post-mortem brain tissue, increased oxidative stress is seen, but as the disease progresses and aggregations of both amyloid and tau expand, the level of oxidative damage appears to decrease [18,135,136]. This has led researchers to suggest that the amyloid protein may have an increased expression in AD because it acts as an antioxidant against mitochondrially induced oxidative stress [16,17,18,137]. Of the different isoforms of the amyloid beta protein, Aβ_1-40_ was shown to have the biggest antioxidant effect, but other isoforms, including Aβ_1-42_, also showed antioxidant properties [137]. This is an interesting theory, as it links several aspects of AD pathology together (mitochondrial dysfunction, oxidative stress and protein aggregation) and gives a physiological role to the amyloid protein, which has, to a certain extent, remained elusive. 

Conversely, oxidative stress has been shown to be highest around AP within the brain of a mouse model of AD [138], although this does potentially contradict the findings from post-mortem human brain studies, showing less oxidative damage in brain areas with high amyloid load [17]. Fragments of the Aβ protein have been shown to have the ability to cause ROS production [139], and both β secretase activity and tau hyperphosphorylation are increased by the activity of ROS [133,139]. This evidence suggests that mitochondrial ROS production may actually exacerbate the accumulation of amyloid and tau aggregates, as opposed to these proteins being produced as antioxidants. The Aβ protein can also interact with a beta-binding alcohol dehydrogenase (ABAD), a dehydrogenase, which has roles in controlling mitochondrial exposure to oxidative stress [134]. Binding of Aβ to ABAD distorts the dehydrogenase’s shape stopping the binding of NAD, and increasing mitochondrial oxidative stress in both mouse models and human neuronal tissue. ABAD would normally bind to CypD, but in the presence of Aβ, this reaction is impaired, which can also lead to an increased MPTP opening in AD neurons, and therefore an increased risk of cellular death cascades being initiated [134,135]. 

Mitochondrial efficiency in AD is likely to be the source of the increased damage seen by the production of ROS. It could be postulated that the AD brain expresses less dismutase enzymes, which are involved in the reduction of ROS. However, work on post-mortem specimens has shown that the expression of these enzymes in AD brains is the same as found in control samples, with differential expression occurring in places of higher oxidative stress [136]. The ETC is the main site of ROS production in the mitochondria, which links the production of ATP with the production of ROS. The altered balance of ROS and oxidative stress within the AD brain further highlights the importance of understanding how the function of the ETC changes in AD. As the Aβ protein has been shown to have potentially both positive and negative effects on the level of ROS production in AD, this could be another example of how the capacity of mitochondrial function is key to the development of AD pathology. The Aβ protein may have an antioxidant role at the start of the disease but, as the capacity of mitochondrial function is already impaired, the antioxidant role of Aβ may not be able to rescue the established mitochondrial AD phenotype, leading to increased Aβ and to the development of more ROS.

## 6. Mitophagy and Cell Death in AD

There is a significant body of literature showing that the dynamics of mitophagy are altered in AD [140,141,142,143,144,145,146,147,148]. Most of these reports focus on the Pink–Parkin mitophagy pathway, but abnormalities in cardiolipin-induced mitophagy have also been reported in AD mouse models [147]. It has been shown in several studies that there is an increased recruitment of Parkin to defective mitochondria in both CNS and peripheral cells in AD, but the mitophagic destruction of these mitochondria is impaired [149,150,151]. As the disease progresses, markers of mitophagy increase in both animal models and post-mortem brain tissue, but the amount of Parkin available for mitophagy is reduced in the cytosol [149]. The reason for the build-up of mitochondria signalled for mitophagy is not fully understood in the current literature, but there is evidence of lysosomal dysfunction (a key part of mitochondrial clearance) in cells that have either the PSEN1 mutations [150], or express the ApoE ε4 allele [152]. The cause for the increased recruitment of Parkin to the mitochondria is not fully understood, but is very dependent on the depolarization of the MMP that, as described above, can be caused by amyloid interacting with the mitochondria. Amyloid also has a propensity to increase ROS production, which may also lead to increased Parkin recruitment, as a further signal to initiate mitophagy. Tau has an effect on mitophagy, but experiments from animal and cell models have shown that tau can both increase Parkin recruitment to mitochondria [141] or stop its translocation from the cytoplasm [142,143]. 

Both ROS production and the dissipation of MMP can be key factors in determining whether a mitochondrion should undergo mitophagy. The function of the mitochondrial permeability transition pore (MPTP), a complex ion pore that is made of a combination of proteins found in the matrix, inner mitochondrial membrane and outer mitochondria membrane [153] also has a key role in determining both the level of cellular mitophagy and cellular death. The function of the MPTP has been shown to be altered in AD, with some studies showing a more constant activation of the pore in cells when compared to controls [151]. The constant opening of the MPTP exposes the cytoplasm to increased ROS and calcium release, which can be signals for mitophagy and cell death. The Aβ protein has been shown to interact directly with the MPTP protein cyclophilin D (CypD), which increases the pores activation [154,155]. There is also evidence that the increased calcium concentration mitochondria are exposed to in AD, due to the increased number of MAM contacts, causes the MPTP to open more frequently to normalize matrix calcium concentration [151]. Inhibiting or knocking down the CypD protein in mouse and cell models, which essentially stops the formation of the MPTP, removes the toxic effect that Aβ has on the mitochondria [154]. This has led to the suggestion that blockage of the MPTP could be used as a therapeutic target in AD, but this negates the fact that MPTP has a physiological role. It is interesting that the blockage of MPTP stops the effect that amyloid has on the mitochondria, but this work has only been performed in mouse models of a relatively young age. It is reasonable to conclude that the blockage of MPTP does not take into account any potential effects that minor changes in mitochondrial function may have on the development of AD, and does not account for the other effects outside of the mitochondria that amyloid has on many different organelles. Cell death can be mediated in many ways in AD via the actions of Aβ. As well as MPTP-mediated cell death, Aβ has a direct effect on caspase signalling, highlighting the complex nature of the disease [156]. Figure 2 summarizes the changes seen to mitochondria in AD.

## 7. Mitochondrial Abnormalities in AD Summary

The sections above highlight the key role that mitochondrial function and homeostasis has in the development and progression of AD. Functional deficits in the ETC lead to a reduction in MMP and ATP generation, which affect the ability of mitochondria to meet the active demands of the cell in AD. There is evidence from multiple studies and multiple cell types that Complex IV has a reduced activity in AD, but functions of the other ETC proteins are also impaired. The pathological process of AD affects the ability of the mitochondria to store and buffer calcium, but it is not yet known if mitochondria have inherent calcium homeostasis deficits independent of the actions of tau and amyloid. AD mitochondria are more likely to produce ROS, which leads to higher levels of cellular and mitochondrial oxidative stress that, as the disease progresses, further disrupts the functions of the ETC. Mitophagy is increased in both the AD brain and peripheral cells at the start of the condition, and becomes more affected as the disease progresses. Mitochondrial dynamics are affected in AD as in mitochondrial biogenesis, with several mitochondrial biogenesis pathways affected.

Much research has focused on mitochondria function in the presence of either amyloid or tau. Less research focuses on whether deficits in mitochondria function are present within a cell without the addition of these pathological aggregates. Having a full understanding of the function of the ETC changes in AD that are independent of the accumulation of amyloid or tau may help identify a group of patients that have a mitochondrial phenotype to the disease. The function of the ETC is fundamental in maintaining calcium, ROS, and mitophagy homeostasis, so a complete understanding of any deficits that occur in ETC function would help to develop our understanding of how these mitochondrial functions are altered when amyloid and tau start to aggregate. Identifying changes in mitochondrial ETC function that are independent of amyloid or tau aggregation requires models of AD that source mitochondria from humans with the condition, and uses model paradigms in which the mitochondria are maintained in a system that is as close to what is seen in vivo as possible. Mitochondrial deficits are seen very early in the course of AD, so a better understanding of how the mitochondrial ETC function predisposes people to the condition may open a pathway to earlier diagnosis and treatment. Table 1 highlights the main changes seen to mitochondrial structure and function in AD. 

## 8. Mitochondrial Dysfunction and Its Current Clinical Imaging Applications

Direct measures of mitochondrial function require laboratory-based analyses of cellular material, which can be sourced via post-mortem or in vivo methodological paradigms. While the histopathological assay of brain tissue at post-mortem is still considered the gold standard for making a definite diagnosis of AD [161], it also offers the opportunity to investigate cellular mechanisms on a region-by-region basis. This is of central importance, since the pathophysiological processes of AD do not affect the brain in a homogeneous manner, but tend instead to progress, showing a degree of preference for certain neural systems, especially in its earliest clinical stages [162,163]. In this respect, the use of neuroimaging is a key methodological resource, as it allows the whole-brain characterization of neural properties and it can also be applied in vivo, with the additional opportunity of monitoring test-retest longitudinal changes.

Two main neuroimaging methods have been devised and used to study brain parameters that act as indirect proxies of mitochondrial function: positron emission tomography (PET) and magnetic resonance spectroscopy [164]. PET is a nuclear medicine method based on the administration of a tracer that binds to a target and is then subjected to a phase of radioactive decay, with the generation of gamma photons. The topographical origin and strength of gamma radiation are then recorded by the instrumentation to reconstruct a 3D map of the (patho)physiological process under investigation.

Moreover, [18F] 2-fluoro-2-deoxy-d-glucose (FDG) is by far the most frequently used PET tracer in the field of AD diagnosis, since it undergoes intracellular decay and allows whole brain visualization of glucose utilization. This is normally done at resting state, when basal metabolic levels tend to be particularly pronounced in the territory of the so-called ‘*default-mode network*’, i.e., including the postero-medial territory of the posterior cingulate cortex and precuneus, the inferior parietal lobe, the medio-prefrontal cortex, the lateral temporal cortex and the hippocampal formation [159]. It is against this baseline configuration that the effect of neurological conditions on glucose metabolism is described. The use of FDG-PET in AD has revealed that patients show reduced metabolism in this territory, in particular in the posterior cingulate cortex and in the inferior parietal cortex [165]. Qualitatively, similar findings emerge when patients with mild cognitive impairment (MCI) are compared with healthy controls [160] and when patients with MCI who convert to dementia are compared to non-converters [166]. Although glucose consumption is tightly coupled to the efficiency of ATP production, other neural mechanisms may, however, contribute to these reductions in AD. First, the amount of ATP required by the brain is proportional to the number of cells [167]. Since the neurodegenerative processes of AD result in cellular loss in the same areas that show hypometabolism, it is possible that reduced resting-state glucose metabolism may at least be due to a reduction in the number of cells (hence, the need to correct for atrophy [168]). A second physiological process showing significant changes in these same areas in AD is blood hypoperfusion [169]. Blood flow and glucose metabolism are tightly coupled in patients with AD [170]. It has thus been proposed that pathology affecting the vascular system (e.g., endothelial and blood–brain barrier dysfunction) results in reduced perfusion, which deprives the neurons from the molecules necessary to generate ATP [171]. As a consequence, although FDG-PET has been widely used as an in vivo marker of cellular metabolism, it is not specific to any specific defined cellular process, such as, for example, mitochondrial integrity.

Magnetic resonance spectroscopy is another technique that allows the in vivo quantification of proxies of mitochondrial function within the brain. Exposed to a magnetic field, various metabolites resonate at different frequencies and generate a distinctive return signal. It is based on this principle that is possible to reconstruct the spectrum of regional expression of a series of molecules, some of which are associated with mitochondrial functioning. This is the case with N-acetylaspartate and lactate, the level of which in posteromedial regions are diminished and raised, respectively, in AD [172,173]. A crucial limitation to the use of this methodology, however, has been the prevalent use of a volume-of-interest approach to establish regional metabolite expression in a hypothesis-based fashion. Whole-brain MRI spectroscopy is still in its infancy and, to our knowledge, only one study has applied this methodology to the study of AD, describing an “FDG-like” pattern of N-acetylaspartate reduction in patients, i.e., in posteromedial, posterior parietal and mediotemporal regions [174].

Over the most recent years, an innovative PET radiotracer compound (18 F-BCPP-EF) has been devised to study mitochondrial function in more detail, as it selectively binds to the mitochondrial complex I [175]. Uptake of this compound shows age association reductions in most areas of the brain, with a particularly steep trend observed in the caudate nucleus [176]. AD was shown to be associated with significantly reduced tracer uptake in the parahippocampal gyrus [177], indicating reduced mitochondrial function in this region. Although limited, this evidence warrants further exploration of the investigative role of this radiotracer, above and beyond the indirect evidence provided by the study of glucose hypometabolism and metabolite spectra.

Additional to these methodologies, a further set of neuroimaging techniques can indirectly help the profiling of the ATP metabolic cascade by measuring whole-brain blood-oxygen utilization. One of the earliest methodologies introduced to measure this neurofunctional property has been the use of PET with an oxygen-labelled tracer, either injected or inhaled. Evidence collected with this methodology indicates that, in patients with dementia due to AD, the rate of oxygen consumption (i.e., ‘*cerebral metabolic rate of oxygen*’) is reduced in temporoparietal regions, while no significant differences are observed in the ‘*oxygen extraction coefficient*’ quantifying the ratio between oxygen consumption and delivery [178]. Moreover, in a study comparing patients with AD and patients with vascular dementia, the former had a reduced cerebral metabolic rate of oxygen in the hippocampus and in the inferior parietal lobe, and reduced oxygen extraction coefficient in the hippocampus and thalamus [179]. This pattern confirms the results obtained with methodologies sensitive to glucose utilization, and indicates that temporal, mediotemporal and parietal regions are particularly susceptible to the metabolic abnormalities observed in AD.

Finally, the use of functional MRI enables the study of voxel-by-voxel oxygenation levels along the axis of time. Of particular interest has been the study of resting-state ‘*functional connectivity*’, a term that indicates the pattern of time-dependent linear association calculated across multiple brain regions. Ample evidence has in fact demonstrated that there is topographical correspondence between pathological mechanisms of neurodegenerative diseases (including AD) and specific neural systems identified with these methodologies [162,180]. One of these systems (mentioned above), the default-mode network, shows reduced functional connectivity in AD in the posterior cingulate cortex and hippocampus [181], suggesting that metabolic dysfunction in these regions not only affects these areas, but also affects the way in which these areas are correlated with one another.

In conclusion, although the field of mitochondrial function imaging is still under development, there seems to be converging preliminary evidence from a spectrum of multiple indirect methodologies, indicating that inferior parietal, posteromedial and mediotemporal regions are particularly vulnerable to metabolic dysfunction. From a methodological standpoint, it is important to consolidate the study of whole-brain patterns of mitochondrial dysfunction in AD to inform and standardise the process of selection of more precise target regions of interest. Region-specific magnetic resonance phosphorous spectroscopy, for instance, has been successfully applied to the study of motor neuron disease [182,183], which is a condition that has the tendency to affect a well-defined target neural system (i.e., the pathways responsible for motor function). This relative selectivity has certainly facilitated the definition of the regions to investigate, in more detail, motor neuron disease. This particular method of imaging can visualize relative amounts of phosphate-containing molecules within the brain and muscle, one of these compounds being ATP. This imaging technique therefore offers another potential mitochondrial imaging target. A comparable effort is thus needed to explore mitochondrial dysfunction in AD using this technique to define the most appropriate regions to be further studied with dedicated experimental paradigms. Table 2 highlights the different imaging techniques discussed and the advantages and disadvantages.

## 9. Mitochondrial Dysfunction: A Future Biomarker of AD?

The development of a disease biomarker starts with identifying a biological parameter which can differentiate between a normal and pathological change in a biological process [184]. Several types of biomarker exist depending on what stage of disease needs tracking. Biomarkers are developed usually to either identify disease, track the response to treatment, or allow physicians to develop a prognosis for a patient with a given underlying condition. As deficits in mitochondrial function are common to several cell types and present in both sporadic and familial AD, there is potential that a mitochondrial biomarker could be developed for AD at any of the three above mentioned stages. In any of the three biomarker types, the approach to developing a mitochondrial biomarker could be done by taking a particular marker of mitochondrial function and investigating if this is affected in large cohorts of AD patients. The imaging of mitochondrial function using functional MRI has shown that mitochondrial dysfunction is already used to differentiate surrogate markers of normal brain mitochondrial function from the abnormalities seen in AD. A recently published study described how fibroblast MSRC may have the potential to be developed into a biomarker of AD, as deficits in MSRC correlate with changes seen in established biomarkers early in the AD disease process [69]. 

When developing a mitochondrial biomarker of AD, identifying a mitochondrial abnormality present in both the CNS and peripheral tissues would be a good starting point. The function of complex IV of the ETC may be a possible future marker, as deficits in the function of this complex are present in CNS and peripheral tissue, suggesting a deficit common to several cell types. This is important when developing a biomarker of AD, as a systemic change in mitochondrial function must reflect a change in the CNS that leads to disease. If a biomarker can be developed through taking peripheral samples, this non-invasive approach would also be beneficial over sampling tissue or fluid from the CNS. A difficultly with using a parameter of mitochondrial function as a biomarker would be the methods of measurement used. Functional assessment of mitochondria currently requires several weeks of processing of biological samples from patients. This would be both expensive and time consuming, making the development of a practical widely available biomarker difficult. If mitochondrial dysfunction did, however, improve the specificity of AD diagnosis, or provide a robust marker of treatment response, then this may be an appropriate time frame to wait. 

These issues could be potentially avoided by measuring protein expression within the blood of a protein marker known to cause the abnormalities in mitochondrial function. Measurement of the NAD+/NADH ratio or Complex II levels may be a useful surrogate marker of MSRC, as Complex II activity has been shown to be directly linked to MSRC [185]. The expression of Complex IV could also be measured, but again, this would not be a direct measurement of mitochondrial function. A metabolomics-based approach to the assessment of the function of the ETC could also be trialled [186]. This method would allow for the measurement of the substrates of the separate complexes, as well as the molecules that are metabolised to create these substrates. This method may also infer mitochondrial functional changes without directly measuring complex function. 

Another approach to develop the changes in mitochondrial function into a future biomarker would be to include mitochondrial assessment as part of a battery of biomarkers for AD. This methodology would include performing multiple tests of mitochondrial function in AD patient cells to define an AD metabolic phenotype. The clinical biomarker system referred to as the “ATN” system has been developed to help characterize people with a diagnosis of AD [187]. This system is based on the presence or absence of amyloid (A), tau (T) or signs of neuronal injury (N) on brain imaging or CSF testing. Since its development, the ATN system has been applied to patients with MCI with the aim of assessing progression to developing AD [188]. By adding mitochondrial functional assessment to the ATN biomarker system, this may increase the reliability of an AD diagnosis at the prodromal or preclinical stage, when therapeutic engagement may be more effective at halting or delaying disease progression. Potentially, adding cellular metabolic assessment to the biomarkers performed on MCI patients may help to predict which patients, and at what time point they will develop dementia. There is already evidence that combining different biomarkers of AD increases the accuracy of clinical diagnosis [189], therefore, the additional assessment of another biological property in people with suspected AD may further improve diagnostic accuracy. 

Understanding the full range of metabolic abnormalities that are common to all cell types in AD may also be a way to develop metabolic functional assessment into a risk stratification tool for people known to be at risk of AD. ApoE ε4/4 genotype [190] and CSF amyloid and tau levels [191,192] can predict, to a certain extent, progression to AD in cognitively normal subjects. Performing a full metabolic assessment of peripheral cells in people who are at risk of AD may again be a way to improve diagnostic accuracy.

Before a peripheral assessment of metabolic function can be developed into a clinically useful biomarker, studies with much larger patient cohorts are needed to validate the findings of the studies described in this review. Very few studies that have investigated mitochondrial function in AD have included large patient cohorts; therefore, further investigations of mitochondrial function on a large scale are needed to identify deficits that are common to the majority of AD patients. The development of improved high throughput screening assays of metabolic functional abnormalities in peripheral cells from patients with AD will allow for large numbers of patients to be screened in this way. A consensus needs to be reached on what an appropriate comparator group might be for a large-scale study investigating the use of mitochondrial function as an AD biomarker. As mentioned in this review article, studies differ in the choice of control groups used, with some researchers preferring to compare mitochondrial changes in AD to other neurodegenerative diseases, as opposed to healthy controls. Mitochondrial functional deficits are common across multiple neurodegenerative disease types [193,194,195,196]. Control groups with other neurodegenerative diseases may, therefore, be the most appropriate reference controls for studies developing and testing an AD specific biomarker. As multiple types of biomarker exist, study control groups could differ depending on the stage of disease that the biomarker is targeted at. 

## 10. Conclusions

Mitochondrial dysfunction is seen early in AD and affects both the CNS and non-nervous system cells and organs. Several abnormalities are present in the function of the mitochondria, especially in the roles regarding ATP production, calcium homeostasis, ROS production, biogenesis and mitophagy. The alterations seen in mitochondrial function may be caused by changes seen to the mitochondrial dynamics in AD, but are also influenced by cellular exposure to both tau and amyloid aggregates seen in AD. Imaging tools already exist that can measure properties of mitochondrial function in the brain indirectly, but this field needs to be developed further. As well as developing imaging biomarkers of AD, there is a potential to turn peripheral mitochondrial functional changes into a metabolic biomarker of the condition. Further work on much larger patient cohorts is, however, needed, including appropriate disease controls. These potentially novel metabolic biomarkers should then be compared with currently available biomarkers and longitudinal sampling to determine if they can be used to provide prognostic or objective monitoring/response to treatments. Future works should focus on deepening the understanding of the mitochondrial structural changes in AD that lead to abnormal mitochondrial function, and to clarify how this knowledge could be developed into future disease modifying therapies for AD and other dementia syndromes sharing similar failing mechanisms.

## Figures and Tables

**Figure 1 biomedicines-09-00063-f001:**
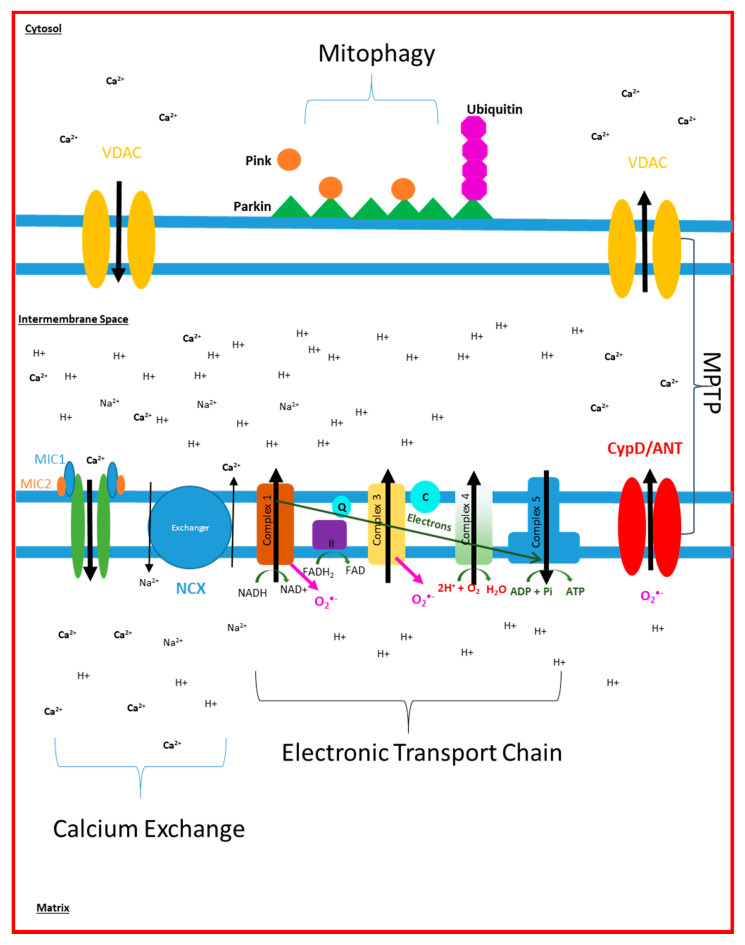
Mitochondrial Function: This figure displays the different elements of mitochondrial function described in this review article. ***Calcium exchange****:* The figure highlights how sodium calcium exchanger (NCX) and mitochondrial calcium uniporter (depicted in green) maintain the matrix concentration of calcium. Calcium enters and leaves the mitochondria via the voltage dependent anion-selective channels (VDAC) found on the outer mitochondrial membrane. ***Electron Transport Chain****:* Complexes 1, 3 and 4 pump protons from the matrix into the intermembrane space which generates a membrane potential that is used by complex 5 (F0F1-ATP Synthase enzyme) to generate ATP. This process consumes oxygen at complex 4 and NADH at complex 1. ***Mitophagy****:* Parkin-dependant mitophagy depends on the recruitment of parkin (green triangles) to the outer mitochondrial membrane, which leads to the recruitment of PINK (orange circles) and eventually ubiquitin (purple octagons) to signal mitochondrial breakdown. ***Mitochondrial Permeability Transition Pore***
*(MPTP):* Is formed of Cyclophilin D (CypD), Adenine Nucleotide Translocator (ANT) and VDAC and controls the movement of calcium and ROS out of the mitochondria.

**Figure 2 biomedicines-09-00063-f002:**
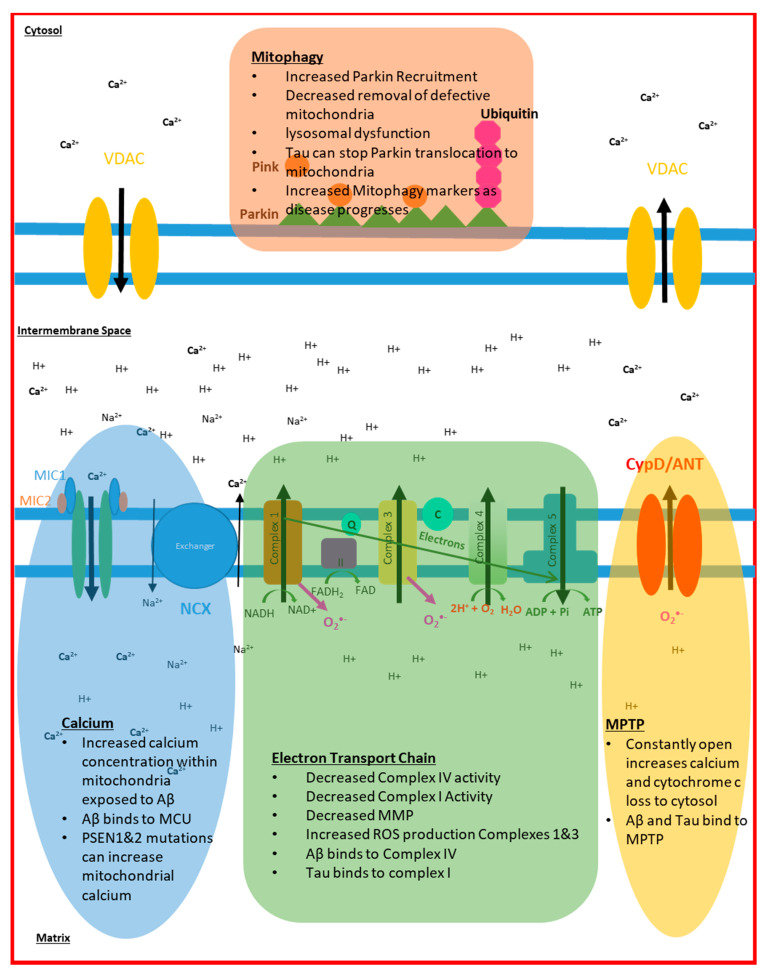
Changes to mitochondrial function seen in Alzheimer’s disease (AD) Highlighted in this figure are the different pathophysiological changes that occur to the mitochondria through the course of AD.

**Table 1 biomedicines-09-00063-t001:** Mitochondrial changes in AD: This table highlights the key papers described in this study and changes to mitochondrial function and form that they describe. Cell type where abnormality has been identified is also highlighted. ↑ indicates an increase, and ↓ indicates a decrease.

Mitochondrial Property	Organ/Cell Type	Change Seen to Mitochondrial Property	Reference
ETC mRNA/Protein Expression	Brain	↑ OxPHOS protein expression ↓ mRNA subunits 1 and 2 Complex IV ↓ RNA of subunit 3 Complex IV ↑ Mitochondrial Subunits of Complex IV and III ↓ Mitochondrial Subunits of Complex I	[157] [158] [30] [33,35,159] [36]
	RBC	↓ OxPHOS genes	[54]
	Platelets	↓ Complex IV subunits	[59,60,61]
ETC Activity	Brain	↓ Complex IV activity	[41,42,43,44,45,46,47,48,49]
	Fibroblasts	↓ Mitochondrial Spare Capacity ↓ OxPHOS, ↓NAD/NADH ratio ↓ Complex IV activity	[66] [68] [67]
	RBC	↓ Oxygen Consumption rates	[106]
	Platelets	↓ Oxygen consumption rates	[110]
Mitochondrial Dynamics	Brain	↓ and ↑ of both fission and fusion proteins ↑ SNO-Drp1	[95,99]
	Fibroblasts	↓ and ↑ of both fission and fusion proteins and ↑Fis1	[27,66] [97,98]
	Blood Cells	↑ SNO-Drp1 in lymphocytes	[96]
Calcium Homeostasis	Brain	↑ IP3R3-VDAC ↑ efflux transporters ↓ influx transporters ↑ Mitochondrial Calcium ↑ MAM Contacts	[124] [124] [123] [116] [116]
	Fibroblasts	↑ MAM Contacts	[123]
ROS Production	Brain	↑ ROS production in areas with lower AP ↑ ROS around AP	[16,138,160] [27,138]
	Fibroblasts	↑ ROS production	[129,130]
	Bloods	↑ ROS production	[137]
Mitophagy	Brain	↑ Parkin Recruitment to mitochondria ↑ Mitochondrial accumulation ↑ Lysosomal dysfunction	[140,141,142,143,144,145,146,147,148] [149,150,151] [150,152]
	Fibroblasts	↑ Parkin Recruitment to mitochondria	[146]

**Table 2 biomedicines-09-00063-t002:** Mitochondrial imaging techniques: This table highlights the different imaging techniques that explore mitochondrial content and function in neurodegenerative diseases. Listed are some of the advantages and disadvantages of each technique.

Imaging Technique	Advantages	Disadvantages
FDG-PET	Technique can be combined with MRI Directly images glucose uptake Can show brain network metabolism in vivo	Uses radiation Resolution of imaging not at a cellular level Not specific to a particular metabolic pathway No temporal information obtainable
18 F-BCPP-EF	Binds to ETC complex I Direct marker of mitochondrial function/content	Uses radiation Resolution of imaging not at a cellular level No temporal information obtainable
MRI Spectroscopy	Can image several different metabolites	Resolution is limited Typically, no whole-brain information is acquired
MRI Phosphorous Spectroscopy	Directly images phosphorus containing compounds such as ATP	Relatively new technique No standardized methodologies developed yet Typically, no whole-brain information is acquired

## Data Availability

No new data were created or analyzed in this study. Data sharing is not applicable to this article.

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
