# Peer review of "Mitochondrial Dysfunction in Alzheimer’s Disease: A Biomarker of the Future?"

_biomedicines, 2021, doi:10.3390/biomedicines9010063_

Round 1
Reviewer 1 Report
Mitochondrial involvement in neurodegenerative disorders such as Alzheimer's disease has been well studied, and this manuscript is a comprehensive review of them. Besides, while the field is still in progress, recent research results on mitochondrial functional imaging for clinical diagnosis are adequately summarized; thus, this manuscript appears to be a pioneering overview in this regard. On the basis of the above, I have judged that this manuscript is worthy of acceptance.
Author Response
We thank the reviewer for their comments.
Reviewer 2 Report
The present review “Mitochondrial Dysfunction in Alzheimer’s disease: a biomarker of the future?” is very well written and explains the potential of mitochondrial function assessment as a tool for Alzheimer’s disease biomarker. The article is very well organized which explains the mitochondrial associated important parameters which could be important biomarker assessment tools. It also describes and compare the aspects of the central nervous system and peripheral system based mitochondrial assessment approaches in the progression of Alzheimer’s disease. It specifically covers in detail the central nervous system based live imaging tools as the potential biomarker approaches keeping in mind the sample limitation with other approaches. Although the present review covers and explains most of the important aspects of the mitochondrial function as a biomarker tool in Alzheimer’s disease, here are a few recommendations to improve the article.
- Mitochondrial dynamics section explains how the mitophagy, fission and fusion gets affected in Alzheimer’s disease. Still one of the important aspect is mitochondrial biogenesis is missing which also determines the pathogenesis and could be a possible explanation for some of the effects mentioned in the section ‘electron transport chain disruption in AD’.
- It’s better to emphasize on peripheral mitochondrial functional assessment as it is getting better with high throughput advancements with Seahorse technology which could make an impact on the development of peripheral biomarker assessment.
- Also metabolomics is one of the important approach to assess the mitochondrial metabolites and correlate with the brain function analysis. It would be important to mention in broad aspects.
- It would be great to include a table with imaging applications with associated pros and cons.
Author Response
We thank the reviewer for their comments and have addressed the edits as suggested below.
The present review “Mitochondrial Dysfunction in Alzheimer’s disease: a biomarker of the future?” is very well written and explains the potential of mitochondrial function assessment as a tool for Alzheimer’s disease biomarker. The article is very well organized which explains the mitochondrial associated important parameters which could be important biomarker assessment tools. It also describes and compare the aspects of the central nervous system and peripheral system based mitochondrial assessment approaches in the progression of Alzheimer’s disease. It specifically covers in detail the central nervous system based live imaging tools as the potential biomarker approaches keeping in mind the sample limitation with other approaches. Although the present review covers and explains most of the important aspects of the mitochondrial function as a biomarker tool in Alzheimer’s disease, here are a few recommendations to improve the article.
- Mitochondrial dynamics section explains how the mitophagy, fission and fusion gets affected in Alzheimer’s disease. Still one of the important aspect is mitochondrial biogenesis is missing which also determines the pathogenesis and could be a possible explanation for some of the effects mentioned in the section ‘electron transport chain disruption in AD’.
- A paragraph on biogenesis has been added in section 4 lines 302-316
- A section has also been added to the graphical abstract, and mitochondrial summary section.
- It’s better to emphasize on peripheral mitochondrial functional assessment as it is getting better with high throughput advancements with Seahorse technology which could make an impact on the development of peripheral biomarker assessment.
- We have now added a sentence to the manuscript describing this lines 658-660
- Also metabolomics is one of the important approach to assess the mitochondrial metabolites and correlate with the brain function analysis. It would be important to mention in broad aspects.
- This has been added in lines 632-636
- It would be great to include a table with imaging applications with associated pros and cons.
- This has been added to the end of section 8, line 600.